# The Effect of Thickness on the Properties of Laser-Deposited NiBSi-WC Coating on a Cu-Cr-Zr Substrate †

**Yury Korobov** [1,2,*]**, Yulia Khudorozhkova** [3]**, Holger Hillig** [4]**, Alexander Vopneruk** [5]**,
Aleksandr Kotelnikov** [5]**, Sergey Burov** [6]**, Prabu Balu** [7]**, Alexey Makarov** [2,8,9]
**and Alexey Chernov** [1]

1   Laboratory of Laser&Plasma Treatment, M.N. Mikheev Institute of Metal Physics of the Ural Branch of the Russian Academy of Sciences, Ekaterinburg 620002, Russia; alexeychernov65@gmail.com
2   Department of Welding Production Technologies, Ural Federal University, Ekaterinburg 620002, Russia; av-mak@yandex.ru
3   Laboratory of Technical Diagnostics, Institute of Engineering Science, Ural Branch of the Russian Academy of Sciences, Ekaterinburg 620049, Russia; khjv@mail.ru
4   Thermal Surface Technology unit, Fraunhofer IWS, Dresden 01277, Germany; Holger.Hillig@iws.fraunhofer.de
5   Research Unit, R&D Enterprise Mashprom, JSC, Ekaterinburg 620012, Russia; rp.mmp@mprom.biz (A.V.); kab@mprom.biz (A.K.)
6   Department of Structural Materials Science, Institute of Engineering Science of the Ural Branch of the Russian Academy of Sciences, Ekaterinburg 620049, Russia; burchitai@mail.ru
7   Research Unit, COHERENT Inc. 5100 Patrick Henry Dr, Santa Clara, CA 95054, USA; prabu.balu@coherent.com
8   Laboratory of Mechanical Properties, M.N. Mikheev Institute of Metal Physics of the Ural Branch of the Russian Academy of Sciences, Ekaterinburg 620108, Russia
9   Laboratory of Materials Science, Institute of Engineering Science of the Ural Branch of the Russian Academy of Sciences, Ekaterinburg 620049, Russia
*   Correspondence: yukorobov@gmail.com
†   This paper is an extended version from our paper, Korobov, Y.; Vopneruk, A.; Kotelnikov, A.; Khudorozhkova, Y.; Burov, S.; Balu, P. Structure analysis of laser deposited NiBSi-WC coatings on a Cu-Cr-Zr substrate. In Proceedings of the the International Conference on Advanced Materials with Hierarchical Structure for New Technologies and Reliable Structures, Tomsk, Russia 9–13 October 2017; https://doi.org/10.1063/1.5013779.

**Abstract:** Ni/60WC coatings on copper substrate were placed via laser deposition (LD). A structural study was conducted using electron microscopy and a microhardness evaluation. Two body abrasive wear tests were conducted with a pin-on-plate reciprocating technique. A tool steel X12MF GOST 5960 (C-Cr-Mo-V 1.6-12-0.5-0.2) with a hardness of 63 HRC was used as a counterpart. The following results were obtained: Precipitation of the secondary carbides takes place in the thicker layers. Their hardness is lower than that of the primary carbides in the deposition (2425 HV vs. 2757 HV) because they mix with the matrix material. In the thin layers, precipitation is restricted due to a higher cooling rate. For both LD coatings, the carbide's hardness increases compared to the initial mono-tungsten carbide (WC)-containing powder (2756 HV vs. 2200 HV). Such a high level of microhardness reflects the combined influence of a low level of thermal destruction of carbides during laser deposition and the formation of a boride-strengthening phase from the matrix powder. The thicker layer showed a higher wear resistance; weight loss was 20% lower. The changes in the thickness of the laser deposited Ni-WC coating altered its structure and wear resistance.

**Keywords:** laser deposition; NiBSi-WC coating; copper; wear

## 1. Introduction

Copper and its alloys are widely used as heat exchangers, conductors, mold plates for continuous casting, and the tuyeres of blast furnaces due to their excellent electric and thermal conductivity. However, in many cases, the material's surface must possess a high level of wear resistance. Therefore, surface engineering, including surface modification and surface repairs, are attractive techniques for improving component performance. For example, electroplating [1], thermal spraying [2], and arc cladding [3] have been adopted. However, the bonding strength between the coating and the copper substrate and/or insufficient wear resistance are critical problems when using these methods.

Laser cladding is useful for producing wear-resistant coating, particularly for copper [4–6] due to the high specific heat input, which is significantly higher than that of arc and plasma treatments.

Laser-deposited metal matrix composites (MMCs) like Ni60WC are characterised by high wear resistance and minimal cracking when applied to steel substrates: This is due to an elastic matrix reinforced with fused tungsten carbides [7,8]. Depending on the rate of heating and cooling, mono-tungsten carbide (WC) and di-tungsten carbide ($W_2C$), characterized by a high level of hardness (16–22 GPa), may be present in the final structure [9]. However, in cases of high thermal loads, these carbides are thermally unstable and may dissolve in the matrix. Consequently, the hard-phase content and wear resistance decreases [10]. Furthermore, compared to steel, heat transfer onto copper is an order of magnitude more intense. This leads to changes in the nature and rate of phase and structural transformations [11]. In addition, the coating is subject to sharp temperature changes, especially at laser-spot boundaries. These can result in a high level of internal stress [12]. The presence of high tensile stress in the coating layer causes sensitivity to cracking [13].

Various modifications to laser surfacing technology are used to solve these problems, in particular preheating [14], feeding additives to the base coating materials [4,15], and deposition of gradient coatings [16]. All of these modifications lead to changes in the heat flow, both in the substrate and in the coating.

The combination of the high heat input of laser radiation and the high heat transfer to the copper makes the dependence of the coating's properties on its thickness significant.

In this case, the heating rate, the volume of the molten metal pool, and the cooling rate change. As we showed earlier, this leads in turn to changes in the structure and properties of the coatings [6]. The evaluation of such changes is important for predicting the properties of the resulting coatings, obtaining input data for modeling processes, and developing surfacing technology. The aim of this study is to assess the effect of changes in the thickness of a laser deposited Ni-WC coating on its structure and wear resistance.

## 2. Materials and Methods

Coupons were prepared from the Cu-Cr-Zr alloy C18150 ASTM (0. 5 Cr, 0.05 Zr, Cu bal) of size $100 \times 100 \times 40$ mm. Often chosen as a continuous casting machine mold material due to its high strength at elevated temperatures, the alloy is characterized by a high recrystallization point: About 550 °C, which is two times higher than that of pure copper [17]. Laser deposition (LD) was performed with a HighLight 10,000D diode laser (COHERENT Inc., Santa Clara, CA-95054, USA) possessing an emission energy of 5000 W, a wavelength of 976 nm, a laser beam of $6 \times 2$ mm, a scanning step of 6 mm, and a powder supply of 36 g/min. The layers were deposited while the substrate was heated up to 200–250 °C. The coating of 0.6 mm thickness (hereinafter LD-thin) was performed with linear scanning velocity of 10 mm/s, area-scanning speed of 72 cm$^2$/s. The coating of 1.6 mm thickness (hereinafter LD-thick) was performed with linear scanning velocity of 2.5 mm/s, area scanning speed of 18 cm$^2$/s. Although thick and thin coatings were deposited with different technological parameters,

the specific heat input is the same. It is proven by uniform melting of the powder of the same feed rate in both cases. Based on this, the comparison between the thin and thick coatings is correct, despite the difference in the layer thickness and the parameters. As seen, the surface square is enough to achieve a stable form of the deposited layer (Figure 1). Thus, one sample was performed for each layer thickness. The feedstock powder, with particle size of 53–150 μm, was a NiBSi-WC mix of 40% Hoganas 1559 and 60% Hoganas 4570.

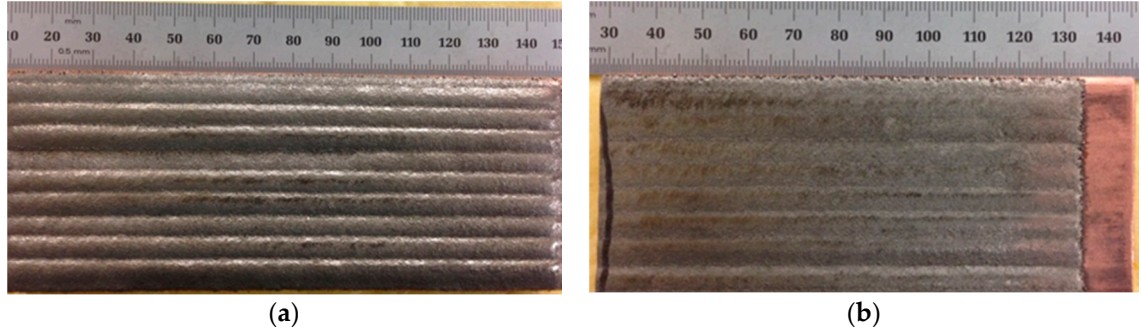

(**a**)　　　　　　　　　　　　　　　　　　　　　　　(**b**)

**Figure 1.** View of the samples as deposited: (**a**) Laser deposition (LD)-thin; (**b**) LD-thick.

Two body dry wear tests were conducted with a pin-on-plate reciprocating technique. The weight loss was determined with the use of balance VLR-200 (measurement error of ±0.12 mg). The tool steel X12MF GOST 5960 (C-Cr-Mo-V 1.6-12-0.5-0.2) was used as the plate material: Its hardness was 63 HRC. Specimens of $7 \times 7 \times 40$ mm cut from the samples were used as pins. The test conditions were the following: A full friction path of 60 m, a pin velocity of 0.08 m/s, a specific load of 6 MPa, and a work surface area of $7 \times 7$ mm.

The structure was studied via scanning electronic microscopy (SEM) with a Tescan VEGA II XMU microscope equipped with wave dispersive (Inca Wave 700) and energy dispersive (INCA Energy 450 XT) microanalyzers. In this work, the EDS method was used. The essence of the method is the registration and subsequent mathematical processing of the characteristic X-ray radiation generated in the studied image by the electron beam. At an accelerating voltage of 15 kV, the generation region ranged from 1 to 2 microns. The standard software of the console performs mathematical processing. The microhardness was evaluated with an HMV-G21 SHIMADZU hardness tester at a 245.2 mN load. The measuring step was 20 μm in the coating and 40 μm in the substrate.

## 3. Results

Characteristic areas of the LD coatings are noted in the structure. In the thin coatings (Figure 2a), these include: 1—Carbide border; 2—inner part of carbide particle; 3—dendrites in the matrix (light area); 4—dark area of the matrix. In the thick coatings (Figure 2b), these include: 1—Small carbide particle; 2—dark area of the matrix, 3—dendrites in the matrix (light area); 4—carbide border; 5—inner part of large carbide particle.

As can be seen, the border of the carbide particles has a coarser structure than that of the inner part of the carbides. This border is characterized by traces of melting and growth from the melted carbide plates (secondary carbides) on the surface of the carbide grains.

There are some differences in the size and structure of carbides in LD coatings of various thicknesses due to differing cooling rates. The LD-thin coating includes only large carbide particles up to 140 μm in size surrounded by the border. The LD-thick coating includes large carbide particles up to 140 μm in size, which are also surrounded by the border and small particles 10–30 μm in size, which is smaller than the diameter interval of the initial carbides. This suggests the precipitation of the tungsten-enriched secondary carbides in the LD-thick coating. The thickness of the carbide border is 3–10 μm on the LD-thick coating. This is two to three times thicker than the LD-thin coating.

A dendritic structure is clearly seen in the matrix of the LD-thick coating. The matrix of the LD-thin coating is much more dispersed.

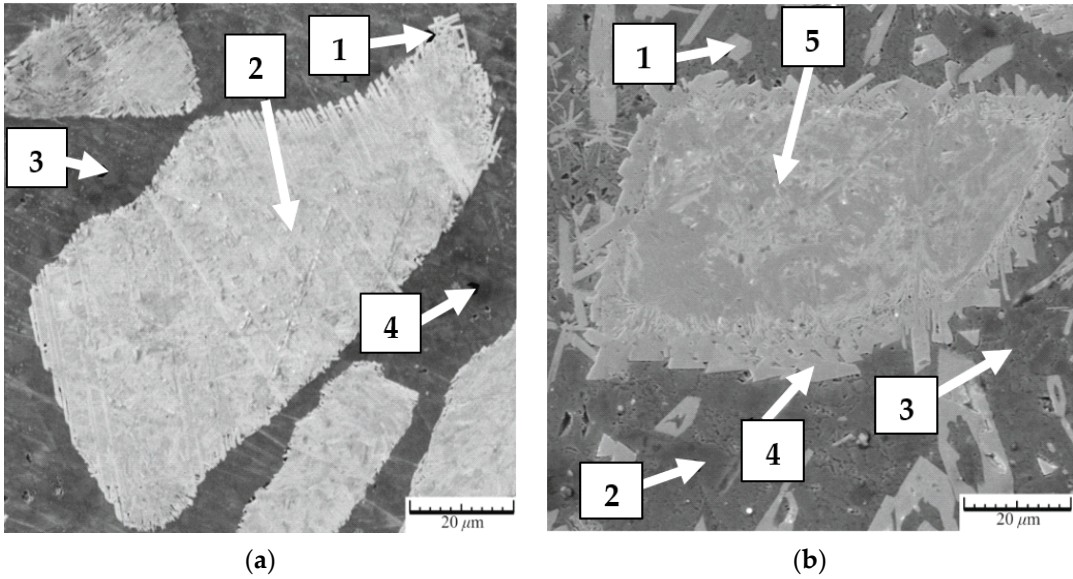

**Figure 2.** SEM image: Carbide particles surrounded by a matrix in LD coating: (**a**) LD-thin; (**b**) LD-thick.

SEM-analysis of the characteristic areas (Table 1, Figures 3 and 4) allows us to state the following. The nickel matrix was enriched by dissolved tungsten carbides. In LD-thin coatings, the precipitation of the secondary carbides is restricted due to the higher cooling rate. In LD-thick coatings, the secondary carbides (point 1) are mixed with matrix material, so the hardness in this zone is lower than in the primary carbides (2425 HV vs. 2757 HV). The dendrites in the LD-thick coatings (point 3) are marked by exhausted boron content compared to the initial Ni-based powder (0.99 vs. 2.9 wt.%). In contrast, the dark areas are enriched with boron (6.6 wt.%). In addition, boron appears in the inner part of the carbide particle and in reprecipitated carbides (1.0–1.3 wt.%). The same picture is seen in LD-thin coatings. This matches the formation of carboborides like W(B,C). As a result, for both LD coatings carbide hardness increases compared to the initial WC powder (2756 HV vs. 2200 HV). In addition, the hardness of the matrix dark areas is three times greater than that of the dendrites.

**Table 1.** SEM analysis of the LD coating (notations correspond to Figure 2).

| Content, wt.% | Characteristic Areas In LD-Thin | | | | Characteristic Areas In LD-Thick | | | | |
|---|---|---|---|---|---|---|---|---|---|
| | 1 | 2 | 3 | 4 | 1 | 2 | 3 | 4 | 5 |
| B% | 1.4 | 1.4 | 6.1 | 5.9 | 1.0 | 6.6 | 1.0 | 1.3 | 1.1 |
| C% | 5.1 | 4.3 | 3.0 | 2.8 | 5.1 | - | - | 4.8 | 3.7 |
| Si% | - | - | 2.1 | 2.1 | - | 3.6 | 1.4 | - | - |
| Fe% | - | - | 0.7 | 0.6 | - | 0.4 | 0.6 | - | - |
| Ni% | 5.3 | - | 76.6 | 76.7 | 2.2 | 89.0 | 71.8 | 15.9 | - |
| Cu% | - | - | - | - | - | - | 4.6 | 1.2 | - |
| W% | 88.2 | 94.4 | 11.5 | 11.9 | 91.7 | - | 20.1 | 76.8 | 95.1 |
| | W(B,C) | W(B,C) | | | W(B,C) | | | W(B,C) | W(B,C) |
| HV | - | 2756 | 667 | 689 | 2425 | 1635 | 212 | 1873 | 2757 |

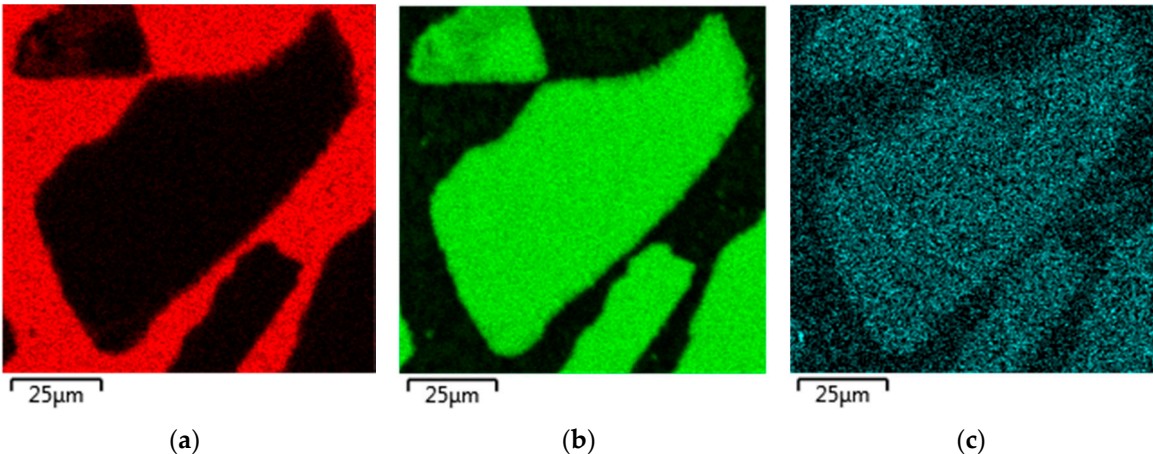

**Figure 3.** Elemental mapping analysis in the characteristic spectra: (**a**) Ni; (**b**) W; (**c**) B. The image corresponds to LD-thin coating, Figure 2a.

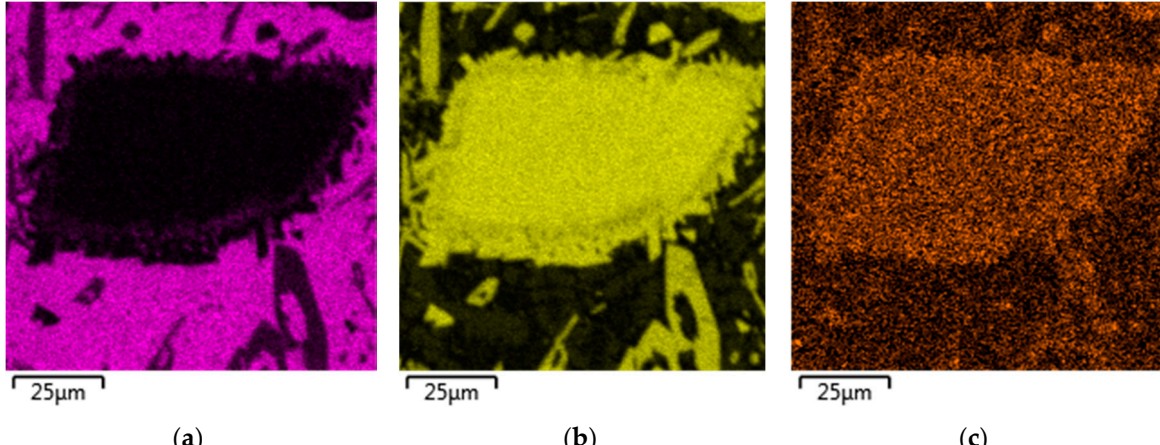

**Figure 4.** Elemental mapping analysis in the characteristic spectra: (**a**) Ni; (**b**) W; (**c**) B. The image corresponds to LD-thick coating, Figure 2b.

Cracks spreading in different directions are present in the coating. They pass along the matrix and cross the carbide particles. The quantity of cracks and their length and width in the LD-thin coating is several times greater compared to the LD-thick coating. This obviously depends on the difference of scanning speed of the surface by the laser beam and the thickness of the coating, which produce a lower level of "thermal shock" in the LD-thick coating.

It should be noted that no cracks were revealed in the secondary carbides. An increase in their share reduced the tendency of the coating to crack. In addition, the rim of the matrix-secondary carbide mixture on primary carbides is a barrier to the propagation of cracks originating in primary carbides during thermal shock (Figure 5). An increase of secondary carbide's share is possible by decreasing the cooling rate of the molten bath and a targeted increase in the number of crystallization centers. A number of technological and metallurgical methods for this are pointed in the introduction [14–16]. The present results can serve as the basis for analysis the indicated technological directions as applied to MMC coatings on copper substrate.

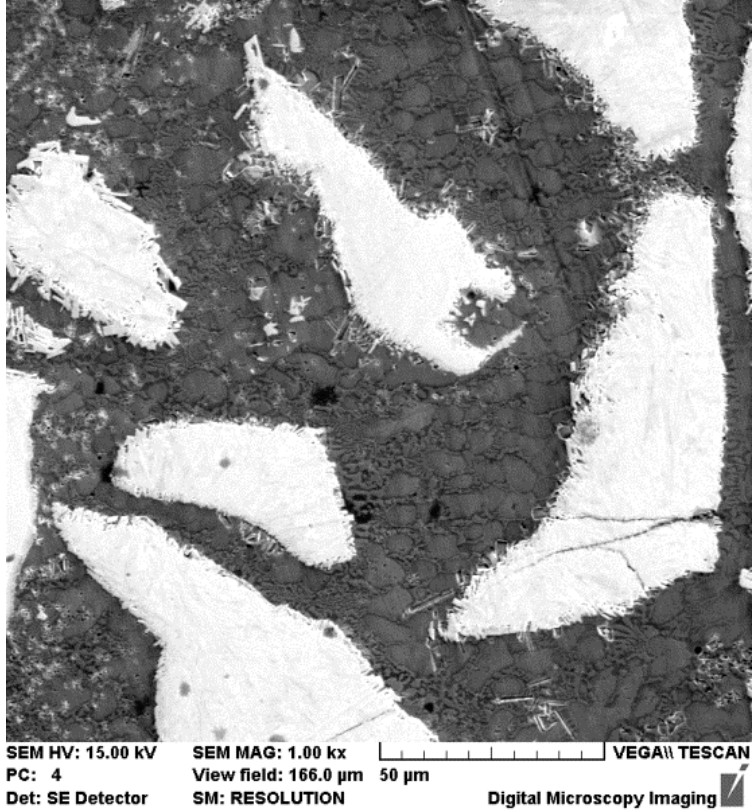

**Figure 5.** A view of crack propagation in carbide particle of LD-thick layer.

Mean microhardness of HV 2700 ± 130 is inherent in carbides both thin and thick coatings. This value corresponds to 90% of the microhardness of the cast tungsten carbides [18], and is twice as high as the microhardness of carbides in HVOF (High Velocity Oxygen Fuel spraying method) coatings [19]. The carbide particles remain unmelted during deposition, which is the main reason of hardness retention. Additional hardness increase is caused by the boride-strengthening phase from the matrix powder. In both cases, the microhardness of the carbide particles does not depend on the location of the particles in the height of the coating: The microhardness is close in value to that on the surface and inside the coating.

EDS line analysis from the surface to the substrate in the LD coatings showed the following (Figure 6). The matrix was mainly formed from nickel. Tungsten concentration peaks are 6%–95% in LD-thin coating and 20%–95% in LD-thick coatings. This confirms widespread precipitation of secondary carbides in the LD-thick coatings. In the LD-thin coating, the mixing zone of Cu and Ni varies from 20 to 40 μm. In the LD-thick coating, the mixing of Cu with the coating is observed at a width of less than 200 μm. This tends to high adhesion strength.

The wear rate of the thick coating is 20% lower than that of the thin one, (weight loss 2.6 vs. 3.2, $\times 10^{-3}$, mg per 1 m of the friction path, correspondingly).

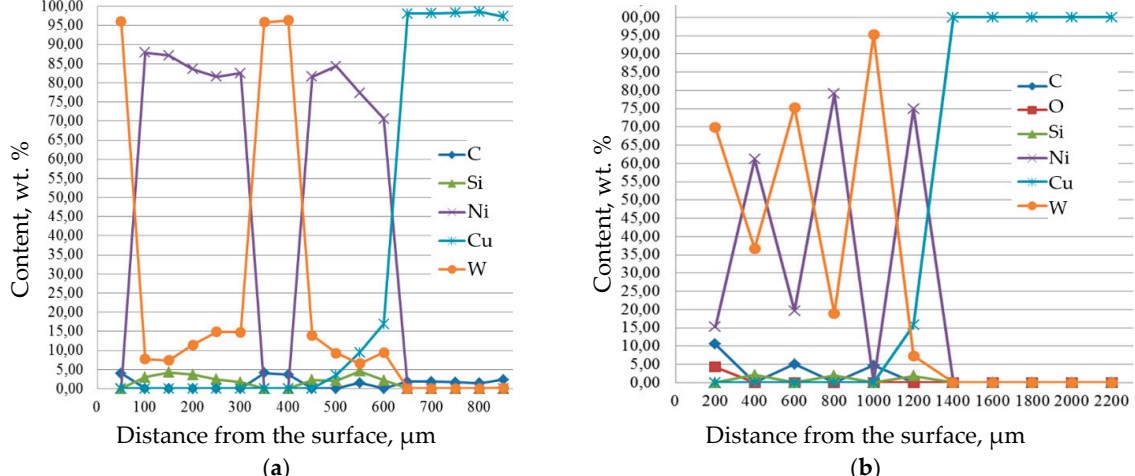

**Figure 6.** The distribution of elements in the cross section of the coating from the surface to the substrate, (**a**) LD-thin; (**b**) LD-thick.

In the SEM analysis of the friction traces, the worn surfaces showed different wear mechanisms (Figure 7). The friction trace after the carbide spots is a transition section combining the matrix and the fine secondary carbides: Plastic indentation is not indicated here. The size of this area was 20–30 microns for both thick and thin layers. This is three to five times larger than the size of the boundary layer around the primary carbides before the wear tests. This indicates the presence of plastic deformation. Plastic deformation occurred on the surfaces of the matrix metal particles, which were broken off from the wear surface. Plastic deformation and micro-cutting were clearly the main mechanisms of dry sliding wear on the matrix surface. The worn surfaces of the carbide spots became comparatively smooth, which indicates that the main wear mechanism in this zone was brittle debonding wear.

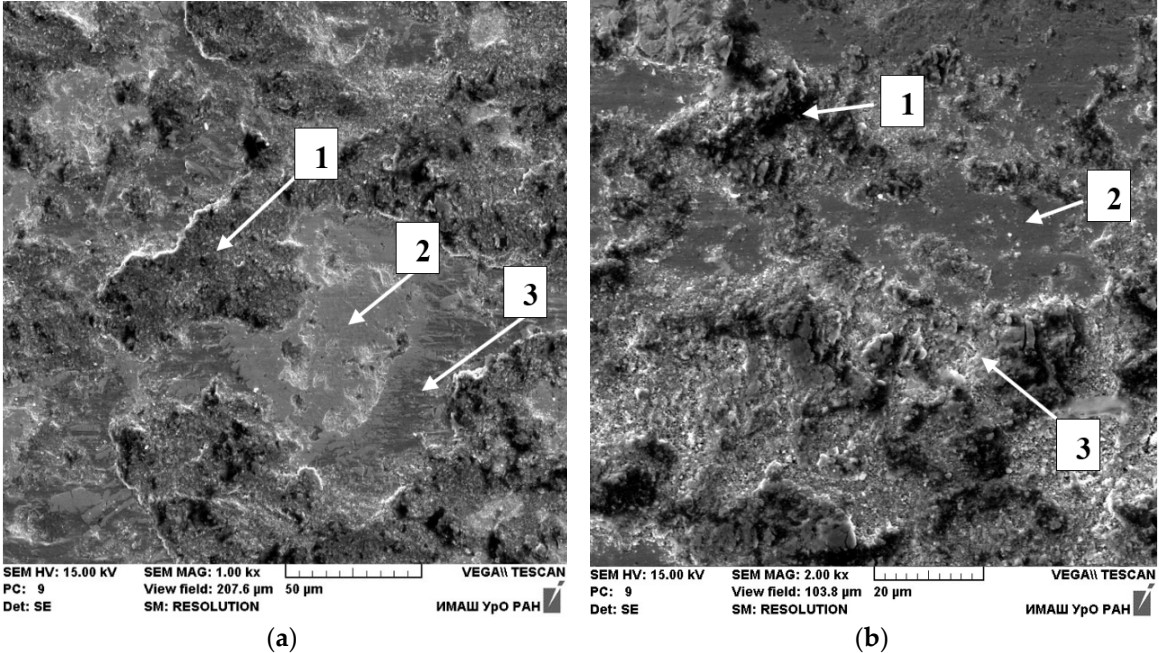

**Figure 7.** Wear surface: (**a**) Thick coating; (**b**) thin coating. (**1**) Matrix; (**2**) supporting sections of carbides; (**3**) transitional section.

The microhardness of similar zones on the worn surfaces varies in the thick and thin coatings (Table 2).

**Table 2.** Microhardness of the worn surfaces.

| Zone | Microhardness, HV0.025 | |
|---|---|---|
| | LD-Thick | LD-Thin |
| Carbides | 2625 | 3200 |
| Matrix | 930 | 880 |
| Transition section | 1260 | 1260 |

The hard carbides had a pinning effect on the metal matrix: Plastic deformation and adhesive wear of the nickel matrix was thus suppressed. The size of the supporting carbide spots in the thick coating was 40–100 microns. This is two to three times larger than in the thin coating. In addition, the supporting spots in the thick coating occupy a larger share of the area than the thin one.

## 4. Discussion

The increased wear resistance of thick coatings can probably be associated with a lower cooling rate, which leads to the following results. Firstly, there is a larger quantity of secondary carbides in thick coatings due to longer precipitation time. Their hardness is two to three times higher than that of a matrix enriched with dissolved carbides. This leads to an increase in the total area of the carbide supporting spots and suppression of matrix wear by the micro-cutting mechanism. Secondly, there is a transition layer at the 'carbide particle–matrix' boundary. This includes the material of the boundary layer between the primary carbide particles and the matrix: The secondary carbides and the matrix were subjected to plastic deformation. The value of the hardness is intermediate between the hardness of carbides and matrix. On the one hand, this contributes to an increase in the area of the supporting carbide spots; on the other hand, it prevents large particles of carbides from spalling. The size of the transition layer is 20–30 microns for both the thick and thin layers. However, its share in the thick layer is smaller due to the greater number of carbides. Reduced wear in the thick coatings can be associated with a favorable ratio of carbides. Its ductility is thus sufficient to fix carbides and reduce spalling. At the same time, a high level of hardness compared to the matrix helps to suppress wear by the micro-cutting mechanism.

Thus, the results of wear tests show that adhesive wear prevails in the matrix while abrasive wear prevails on the carbide particles.

## 5. Conclusions

- The uniform deposition of the 0.6–1.6 mm thick NiBSi-WC layer on the Cu-Cr-Zr substrate can be accomplished using a diode laser possessing the emission energy of 5000 W, the wavelength of 976 nm, when the substrate is pre-heated to 200–250 °C.
- The carbides in the laser-deposited layers are characteristic areas, which are more susceptible to cracking. This susceptibility increases as the layer thickness decreases. Optimization of rate of scanning, the form of the laser beam, and the size of the overlap zone of passes are required to obtain a crack-free layer.
- The size of the Cu–Ni mixing zone during laser processing varies from 20 to 40 μm in thin coatings. In thick coatings, the mixture of Cu with Ni is observed at a width less than 200 μm. This indicates high adhesion strength.
- Precipitation of secondary carbides takes place in the thicker layers. Their hardness is lower than that of primary carbides in the deposition (2425 HV vs. 2757 HV) because they mix with the matrix material. In the thin layers, the precipitation is restricted due to a higher cooling rate.
- The microhardness value of laser-deposited tungsten carbides is 0.9 that of the cast carbides: It is about two times higher than the carbides in HVOF coatings. For both LD coatings, carbide hardness increases compared to the initial WC-containing powder (2756 HV vs. 2200 HV). Such a

high level of microhardness reflects the combined influence of low thermal destruction of carbides during laser deposition and the formation of a boride-strengthening phase from the matrix powder.

- Two body dry wear tests showed that weight loss is 20% less for the LD-thick layer. This is due to the higher cooling rate of the LD-thin layer, which leads to a transition from adhesive wear in the thin layers to abrasive wear in the thick layers. This is due to the following reasons:

  – The enlargement of the square of the supporting contact patch with reprecipitated carbides;
  – The transitional layer on the carbide–matrix interface is intermediate in terms of hardness (1900 HV) and plasticity. In the 1.6 mm layer, its thickness is two to three times higher. On the one hand, this contributes to an increase in the patch square; on the other hand, it keeps the carbide particles from spalling.

**Author Contributions:** Y.K. (Yury Korobov) and A.V. conceived and developed the experiments; Y.K. (Yulia Khudorozhkova) and P.B. performed the experiments; S.B., A.K. and A.M. analyzed the data; A.C. and H.H. provided the reagents/materials/analysis tools. All authors contributed to the preparation of the manuscript.

**Funding:** This research received no external funding.

**Acknowledgments:** This work was supported by the state orders of IMP UB RAS on the subjects "Laser", "Structure". The authors are grateful to the collective use center "Plastometry" of the Institute of Mechanical Engineering of the Ural Branch of the Russian Academy of Sciences for their help in conducting structural studies.

**Conflicts of Interest:** The authors declare no conflict of interest. The founding sponsors had no role in the design of the study, in the collection, analyses, or interpretation of data, in the writing of the manuscript, or in the decision to publish the results.

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
