# Peer review of "The Effect of Thickness on the Properties of Laser-Deposited NiBSi-WC Coating on a Cu-Cr-Zr Substrate†"

_photonics, doi:10.3390/photonics6040127_

Round 1

Reviewer 1 Report

In this manuscript, the authors presented the results of the investigation of NiBSi + WC coatings deposited on the copper alloy by the use of the laser technique.

Dear authors, you did a great effort, but some improvements are still possible. Therefore, please take into the consideration the following comments and suggestions ordered according to the sections of the paper.

ABSTRACT

The authors should modify the Abstract. I suggest omitting the first two paragraphs, because the general statements and the author’s goals are mixed. In fact, it is not clear when something is generally stated or is about author’s study (especially for the second sentence).

Furthermore, some conclusions in the Abstract are twice mentioned (“The thicker layer showed a higher level of wear resistance; the weight loss was 20% less”. “The thicker layer showed a higher wear resistance; weight loss was 20% lower.”).

In addition, the cold work tool steel is designated in two different ways (in the Abstract - AISI D2, and in the section Materials and Methods - according to the GOST norm). The hardness of this steel was stated as 63 HRC (in the Abstract) and in the section Materials and Methods, only 60 HRC. What is the right value? The difference is not negligible.

The abbreviation LD, not previously explained, appears in the Abstract.

The authors stated: “Laser cladding is an attractive method for producing Ni-WC coatings, particularly on copper“. The authors should proove this statement by citing more papers (in the Introduction) dealing with depositing the NiBSi+WC coatings on copper alloys by the use of laser technique.

INTRODUCTION

The literature review is too general. Some critical insight should be provided and a bit detailed description of the cited paper could be done. For example, it would be very interesting to comment two opposite results (“Laser-deposited MMCs like Ni60WC are characterised by high wear resistance and minimal cracking when applied to steel substrates: …“ and „The presence of high tensile stress in the coating layer causes sensitivity to cracking [10].“). Please, take it into the consideration.

The abbreviation MMC (metal matrix composite?) should be explained at the first appearance. The same is valid for CCM (in the section Materials and Methods).

MATERIALS AND METHODS

The term “designed experiments” was mentioned in the Author’s contribution. This term includes varying the input variables in a systematic way to investigate the influence on the output variable (response); the statistical analysis of the obtained data is obligatory. I think that the design of experiment approach is not applied in the present study.

Furthermore, how many samples was deposited for each thickness? It is a very fast process and we need to ensure the repeatability of the results.

Is it possible (or right) to compare the thick and thin coatings obtained with different parameters (Table1)? Did different parameters have to be used? If possible, it would be better to have the same parameters or combine different thicknesses and laser parameters, i.e. to perform designed experiment.

Is it about 5 kW (“emission energy of 5000 W“ - Materials and Methods) or 10 kW laser (Conclusion)?

The microhardness – load 245.2 mN? (Materials and Methods). What is about HV0.2 (0,2*9,81=1,961 N) (Results). Which load was applied?

The chemical composition of the substrate material should be provided.

RESULTS

Figure 2b is of different size than Figure 2a (unclear letters).

When explaining the Figure 2, it should differentiate between 2a and 2b.

Unclear sentence: „That the carbides sank quickly may have been caused by the double difference in density between the nickel matrix and the tungsten carbide.“

Line 123 (point 1 instead of point 5)

The authors stated: “Cracks spreading in different directions are present in the coating. They pass along the matrix and cross the carbide particles. The quantity of cracks and their length and width in the LD thin coating are several times greater compared to the LD thick coating. This obviously depends on the  difference of scanning speed of the surface by the laser beam and the thickness of the coating, which produce a lower level of "thermal shock" in the LD thick coating.” The authors should explain which parameters would probably give coatings without cracks. Are the results of the present investigation (because of the cracks) are applicable in the practice?

The authors concluded as follows: “These values correspond to 90% of the microhardness of the cast tungsten carbides [15], and are twice as high as the microhardness of carbides in HVOF coatings [16].“ This conclusion maybe can be written here (but still, can laser and high velocity deposition be compared?). I think that it is not necessary to mention it in the Abstract and Conclusion because in these sections it seems to be about the author’s own researches of HVOF (or HVAF) and comparisons.  

Regarding the mapping EDS analysis, the images should be provided.

Substrate or base material – please use this term uniformly (as well as chipping and spalling).

The authors should discuss the coating/substrate interface.

Author Response

Dear Sir,

Thank you for the review. The answers are enclosed.

Sincerely.

Yu. Korobov

Reviewer 2 Report

The authors prepared two sets of a metal matrix composite, having a composition of NiBSi-WC, coated on a Cu-Cr-Zr alloy substrate by laser-deposition, in which each set had a distinct thickness. The materials were studied by electron microscopy, and microhardness testing and conclusions were drawn from the degree of mixed components and film thickness. Although the topic is interesting, I do not recommend publication without a major revision of the manuscript. These are the major components to be addressed:

(i) The paper has a large amount of text overlapping with another published work: Yury Korobov et al., Structure Analysis of Laser Deposited NiBSi-WC Coatings on a Cu-Cr-Zr Substrate, AIP Conference Proceedings 1909(1):020098 (DOI: 10.1063/1.5013779). This is characterized as self-plagiarism, and authors are advised to be very careful with that as it can lead to mitigation by many journals. In addition, the conclusions are almost identical between published work and this paper. Authors should clearly state the novelty and cite the previous work as well.

(ii) A large portion of the discussion relays on the chemical analysis in which the technique performed was not well described. The analysis was performed by SEM, but it is unclear if the analytical technique was WDS, EDS, or even PIXE. The authors should describe the analytical method better. The most common is the EDS, and I will assume this technique forward; however, the discussion is valid for other techniques too. First, there are too many significant figures, considering the precision of this technique. If the EDS is well-calibrated (which can be challenging), the SEM-EDS will provide a precision of about 0.1 wt%. Thereby, the authors should provide one digit after the decimal instead of two. Why is this important? Some of the regions present in Figure 3 were considered different in composition, but the values are two close, and the difference may represent a statistical error. Therefore, there is a chance of the regions having a statistically similar composition. Note that light elements, e.g. B, may present precision > 0.1%.

(iii) Line 138 to 140. The authors affirm that value A (HV 2700 ± 130) exceeds value B (HV 2680 ± 130). Considering the uncertainty, both values are statically the same, and conclusions cannot be drawn from this data.

(iv) Figure 4 is unnecessary and can be transferred to text only. Please provide uncertainties to prove that both measurements are statistically different. This is important when values are close.

Author Response

(The authors gave the same response as above.)

Round 2

Reviewer 1 Report

Dear authors, first, I want to suggest to you avoiding the use of “Dear Sir”, since the reviewer can be a woman too.

In addition, the responses to the reviewer should be more detailed or the changes should be written in a different color in revised version.

Dear authors, please take into the consideration the following comments and suggestions ordered according to the sections of the paper.

ABSTRACT

 “…evaluation.” instead of “…evaluation .”

The suggestion – “The following results were obtained:” (instead of “found”).

coating or layer – please use this term uniformly (where applicable).

Keywords – NiBSi instead of NiBSI

INTRODUCTION

The added reference 18 should be numbered and listed as 6? I think that it is not possible to have, 4, 5 and then 18. It is not common. Furthermore, the reference 18 is not a newly added. Instead of that, there are some newly added references in revised version, which would be more appropriate to cite here instead of reference 18. In that way, the authors will fullfill my expectation about the approval of frequent investigation of laser cladding to copper alloys.

MATERIALS AND METHODS

I think that the explanation „Although thick and thin coatings were deposited with different parameters, they can be compared, since the powder supply in both cases, which characterizes the heat input, is the same.“ is not scientifically justified since it is not the only parameter influencing the heat input.

I agree withe the rest of the statement: „As seen, the square is enough to achieve a stable form of the deposited layer (Fig. 1).“. But, in the future, please use more samples (replicates). Maybe the term “sample” can be used instead of square or coupons

“deposited” instead of “depositted “XMU microscope“ instead of „XMU microscope.“

 Lines 103 – 107: “Fig. 2” instead of “Fig. 1”

Two different chemical compositions for the substrate are given in revised version and in Responses to reviewer.

- bronze or copper substrate, please use uniformly

The list of references should be edited – there are too many errors.

Author Response

Dear colleague,

Thank you for a thorough and in-depth review,

Sincerely,

Iurii Korobov PhD (Welding and related technologies),
Principal Research Scientist,
M.N. Mikheev Institute of Metal Physics of the Ural Branch of the Russian Academy of Sciences, (http://www.imp.uran.ru/?q=en/) Professor, Welding Technology Department, Institute of New materials and Technologies of Ural Federal University (http://urfu.ru/en/home/) +7 9193792016 yukorobov@gmail.com

Reviewer 2 Report

The authors have addressed all comments previously raised, and paper quality has improved. Therefore, I recommend the publication of this paper.

Author Response

(The authors gave the same response as above.)
